# Genome-Wide Identification, Expression Pattern and Sequence Variation Analysis of *SnRK* Family Genes in Barley

**DOI:** 10.3390/plants11070975

**Published:** 2022-04-03

**Authors:** Jiangyan Xiong, Danyi Chen, Tingting Su, Qiufang Shen, Dezhi Wu, Guoping Zhang

**Affiliations:** 1Department of Agronomy, Key Laboratory of Crop Germplasm Resource of Zhejiang Province, Zhejiang University, Hangzhou 310058, China; 11916033@zju.edu.cn (J.X.); 22016182@zju.edu.cn (D.C.); sutingting@zju.edu.cn (T.S.); shengqf@zju.edu.cn (Q.S.); 2College of Agronomy, Hunan Agricultural University, Changsha 410128, China; 3Linyi Institute of Agricultural Sciences, Zhejiang University, Linyi 276000, China

**Keywords:** *SnRK*, barley, gene structure, expression pattern, SNP distribution

## Abstract

Sucrose non-fermenting 1 (SNF1)-related protein kinase (SnRK) is a large family of protein kinases that play a significant role in plant stress responses. Although intensive studies have been conducted on *SnRK* members in some crops, little is known about the *SnRK* in barley. Using phylogenetic and conserved motif analyses, we discovered 46 SnRK members scattered across barley’s 7 chromosomes and classified them into 3 sub-families. The gene structures of *HvSnRKs* showed the divergence among three subfamilies. Gene duplication and synteny analyses on the genomes of barley and rice revealed the evolutionary features of *HvSnRKs*. The promoter regions of *HvSnRK* family genes contained many ABRE, MBS and LTR elements responding to abiotic stresses, and their expression patterns varied with different plant tissues and abiotic stresses. *HvSnRKs* could interact with the components of ABA signaling pathway to respond to abiotic stress. Moreover, the haplotypes of *HvSnRK2.5* closely associated with drought tolerance were detected in a barley core collection. The current results could be helpful for further exploration of the *HvSnRK* genes responding to abiotic stress tolerance in barley.

## 1. Introduction

Under natural conditions, plants are constantly exposed to a variety of abiotic stresses, such as drought, salinity and extreme temperatures, resulting in the inhibition of growth and development. In responding to these abiotic stresses, plants have evolved a complex genetically regulated system of tolerance or adaptation [1]. For example, barley (*Hordeum vulgare*) could rapidly adapt its enzyme system to scavenge reactive oxygen species (ROS) when exposed to salt stress [2,3]. However, the genetic mechanisms of the tolerance to abiotic stresses in plants are still not fully understood.

Protein kinases and phosphatase-mediated signal transduction are important regulators involved in the genetic networks responding to environmental changes or abiotic stresses [4]. Sucrose non-fermenting 1 (SNF1)-related protein kinase (SnRK) is a protein kinase family involved biochemical response to stresses in plants [5]. All SnRK proteins share a conserved serine/threonine protein kinase domain in the N-terminal, while they are variable in the C-terminal. The SnRK proteins could be grouped into three subfamilies according to the structures of the C-terminal [6]. The SnRK1 subfamily consists of three domains, including a kinase domain in the N-terminal, an ubiquitin-associated (UBA) domain and a kinase-associated 1 (KA1) domain in the C-terminal, which is the homologues of SNF1 in yeasts and AMP-activated protein kinases (AMPK) in animals [7]. The SnRK2 subfamily harbors the regulatory C-terminal domain containing acidic amino acids, either Glu or Asp [8]. The SnRK3 subfamily contains two conserved domains in the C-terminal, including NAF (named with conserved amino acids N, A and F) and PPI (Protein–protein interacting) domains [9].

It is well documented that the SnRK1 protein kinase is a core regulator in plants to sense energy deficiency [10]. SnRK1 can phosphorylate and inhibit four important enzymes in plant metabolisms, including HMG-CoA reductase (HMGR), sucrose phosphate synthase (SPS), nitrate reductase (NR) and seaweed phosphate synthase 5 (TPS5). SnRK1 regulates the metabolism and development of plants through the above four enzymes [11,12,13]. SnRK2 plays a key role in responding to ABA and salt stress. In *Arabidopsis* and rice, all member genes of the *SnRK2s* (*AtSnRK2.1-2.10* and *OsSAPK1-10*) could be activated by ABA and salt stresses [14,15]. For instance, *AtSnRK2.6*, also designated as *Open stomata 1*, increased the expression level by ABA in guard cells [16]. *AtSnRK2.4* and *AtSnRK2.10* respond to salt stress by regulating the expression levels of several genes for ROS generation and removal [17]. In rice, *OsSAPK8* is a key positive regulator when exposed to salt, drought and chilling stresses [18]. SnRK3, named as CIPKs (CBL-interacting protein kinases), regulates the transports of Na^+^, K^+^ and NO_3_^−^ by combining with Ca^2+^-dependent CBL to regulate downstream genes and enhance abiotic stress tolerance in plants [19,20,21]. For example, SOS2 (CIPK24) protein can interact with SOS3 (CBL4) to regulate the activity of plasma membrane H^+^/Na^+^ antiporter (SOS1) in a Ca^2+^-dependent way in *Arabidopsis* [22,23]. SnRK3.17/SnRK3.12/SnRK3.23 proteins interact with CBL2/3 at the vacuole membrane, co-chelating Mg^2+^ in the vacuole [24]. In addition, CIPK18 and CBL3 are essential for the functions of NHX5 and NHX6 in maintaining Li^+^ homeostasis [25]. Apart from salt stress, a great number of CIPKs were also induced by mild freezing shock treatment [26]. In short, because the *SnRK* gene family is so vast and plays such an important role in responding to abiotic stresses, it is imperative to identify *SnRK* genes and reveal their roles.

Gene families are widespread in plants, and there are the structural and functional similarities among them. In addition, the duplicated family gene pairs among the different genomes can be used to identify the conserved functional genes and determine species relationships, which are beneficial for understanding the mechanisms of gene family evolution and functional differentiation [27]. Recently, many reference genomes of plants are sequenced and assembled, and a variety of *SnRK* family genes have been identified through bioinformatics analysis. However, similar studies are rarely performed on barley. Accordingly, the current study was carried out to identify *HvSnRK* genes and reveal their evolution mechanism, so as to understand the expression patterns of the genes in responding to abiotic stresses in barley.

## 2. Results

### 2.1. Identification and Phylogenetic Analysis of SnRKs in Barley

Based on the query protein sequences from *Arabidopsis* and rice (Appendix A), totally 46 proteins were identified as the SnRK members in barley (Appendix A). The 46 HvSnRK proteins consist of 341 to 797 amino acids, with a molecular weight of 38.5 to 89.9 kDa.

To verify the evolutionary relationship of SnRK proteins among diverse plant species, we constructed a phylogenetic tree utilizing 48, 39 and 46 SnRKs from rice, *Arabidopsis* and barley, respectively (Figure 1). The 133 SnRK proteins were clustered into 3 groups, as found previously in *Arabidopsis* [6]. In barley, 4 proteins with Pkinase (PF00069), UBA (PF00627) and KA1(PF02149) domains are clustered into the HvSnRK1 subfamily; 11 proteins belonged to the HvSnRK2 subfamily with high similarity to the AtSnRK2 and OsSnRK2 subfamilies; and another 31 proteins with Pkinase and NAF (PF03822) domains were grouped into the HvSnRK3 subfamily (Figure 2A). Subcellular localization prediction indicated that HvSnRK1s were mainly located in cytoplasm, HvSnRK2s were mainly located in nuclear and HvSnRK3s were mainly located in the plasma membrane (Appendix A), suggesting that the three subfamilies of HvSnRKs may have the different biological functions.

### 2.2. Gene Structures, Protein Motifs and 3D Structures of HvSnRKs

The gene structures of *HvSnRK* in the same subfamily are similar. Four genes of the *SnRK1* subfamily contain eleven exons. Meanwhile, 10 out of 11 members in the *HvSnRK2* subfamily contain 8 or 9 exons, and only *HvSnRK2*.6 contains 3 exons. The *HvSnRK3* subfamily genes have a large variation in exon amount, ranging from 1 to 16 (Figure 2B). In addition, *HvSnRK3s* can be divided to two subgroups, while one subgroup member has at least eight exons and the other group members have no more than four exons, except *HvSnRK3.26*, which contains eight exons (Figure 2B, Appendix A).

The protein sequences of 46 HvSnRKs were analyzed, and 10 conserved motifs were obtained (Figure 2C). The motifs 1/2/3/5/6 are related to the conserved domains of phosphokinase, while motif 9 is related to the NAF domain (Appendix A). The conserved motif analysis reveals that identical conserved motifs are found in the same subfamily. For example, all members in the HvSnRK1s contain the motifs 1/2/3/4/5/7/10, and the motifs 1/2/3/4/5/6/7 exist in the HvSnRK2 subfamily, while the motifs 1/2/3/4/5/7/8/9/10 exist in the HvSnRK3 subfamily. These results indicate that the same subfamily of HvSnRK has high similarity in the gene structure and amino acid sequence. It can be seen from 3D structures of HvSnRK proteins that three subfamily proteins have the similar 3D structures in the N terminal but differ in the C terminal (Figure 3).

### 2.3. Chromosomal Location and Gene Duplication of HvSnRKs

In order to reveal the evolution of the barley *SnRK* genes, the chromosome location of the *SnRK* genes was determined (Figure 4). *HvSnRK1* subfamily genes were distributed in chromosomes 1, 3 and 4; *HvSnRK2* subfamily genes are found on chromosomes 1, 2, 3, and 4; and *HvSnRK3* subfamily genes distributed over all chromosomes. Interestingly, *HvSnRK3.12* and *HvSnRK3.13* were clustered within 10 kb in chromosome 3. 

In the colinear segment of the barley genome, six *HvSnRKs* gene pairs were discovered, with the two genes of each pair being located on the distinct chromosomes. Moreover, the collinearity of the *SnRK* family genes between the genomes of barley and rice was also analyzed, with 33 pairs of *SnRK* genes being detected (Figure 5B and Appendix A).

### 2.4. Stress-Related Cis-Elements in the Promoters of HvSnRK Genes

For determining the expression pattern of *HvSnRK* genes, 2 kb promoter sequences of all *HvSnRK* genes were extracted from the barley genome database to analyze the *cis*-acting elements. We analyzed the ABA-signal-related components ABRE (ACGTGG/TC), drought-response components MBS (CAACTG) and low-temperature-related components LTR (CCGAAA) (Figure 6, Appendix A). As a result, we found 43 of all 46 *HvSnRK*s, except *HvSnRK3.12*, *HvSnRK3.13* and *HvSnRK3.28,* contained the *cis*-elements responding to the 3 abiotic stresses. Among the 43 *HvSnRKs*, 42 genes had ABRE, 19 genes had MBS and 26 genes had LTR. It is indicated that many ABRE, MBS and LTR elements were identified in the promoter of *HvSnRK* family genes, which may respond to abiotic stresses by modulating gene expression.

### 2.5. Expression Profiles of HvSnRKs in Different Tissues and under Different Abiotic Stresses

All *HvSnRKs* were expressed at various levels in 15 tissues of the barley cultivar Morex, but there was no expression in the developing young inflorescences (Figure 7A and Appendix A). According to the expression levels in different tissues, *HvSnRKs* can be divided into three groups. Group-1 consists of 18 genes with high expression level, including 3 *HvSnRK1s*, 6 *HvSnRK2s* and 8 *HvSnRK3s*. Group-2 consists of 17 genes with moderate expression level, including 2 *HvSnRK1s*, 3 *HvSnRK2s* and 12 *HvSnRK3s*. Group-3 had 13 genes with the low expression level, including 1 *HvSnRK1s*, 2 *HvSnRK2s* and 10 *HvSnRK3s*. These results indicate the dramatic difference in the expression level among *HvSnRK* genes.

In order to understand the expression profiles of the barley *SnRK* family genes in response to abiotic stresses, we integrated the transcriptome data of barley *SnRK* genes responding to abiotic stresses including drought [28], high salt [29], high temperature [30] and waterlogging [31]. On the whole, most *HvSnRK* genes showed a significant change in expression levels under various abiotic stresses, and a few of them showed little change (Figure 7B, Appendix A). For example, *HvSnRK2.5* showed increased expression under drought and salt stresses, while *HvSnRK2.10* had little change under the two stresses. From the expression profiles of *HvSnRK* genes under abiotic stresses, it can be seen that the different *HvSnRK* subfamilies are not consistent in their functions.

### 2.6. Functions and Regulatory Networks of HvSnRKs

The protein–protein interaction (PPI) was predicted to further understand the biological roles and regulatory networks of HvSnRKs. A total of 20 functional proteins that interact with HvSnRK proteins were discovered (Figure 8). Most of the proteins that interacted with HvSnRKs were functionally confirmed ABA signaling components, such as PP2C and PYR/PYL (Figure 8). It was also discovered that the majority of SnRK3 proteins interacted with Calcineurin B-like proteins (CBLs). These SnRKs interaction proteins can be grouped into four categories according to their function annotation, including phosphatase 2C family proteins (ABI5, PP2C6, PP2C9, PP2C30 and PP2C50), ion channel proteins (CBL1, CBL4, CBL6 CBL8 and AKT1), PYR/PYL proteins (PYR1, PYL1, PYL2, PYL4, PYL5, PYL8 and PYL9) and stress- or development-related transcription factors (bZIP27, VP1 and TRAB1).

### 2.7. Sequence Variation of HvSnRKs in a Barley Core Collection

The SNPs of *HvSnRK*s from 100 barley core accessions were detected. On average, a *SnRK* gene contained 10 SNPs (Appendix A). However, the SNP density of various subfamilies differed greatly, with the averages of the 3 subfamilies being 7.0, 14.5 and 9.2 SNPs/gene, respectively. Moreover, the SNP density of each *HvSnRK* gene within the same subfamily also showed the obvious difference. Thus, no SNP was detected for *HvSnRK1.4*, while there were 15 SNPs for both *HvSnRK2.5* and *HvSnRK3.24*. Here, the detailed SNP distribution of *HvSnRK2.5* and *HvSnRK3*.24 was shown in Figure 9. For *HvSnRK2.5*, 2, 10 and 3 SNPs were detected in the promoter, exon/intron region and 3′UTR regions, respectively (Figure 9A). For *HvSnRK3.24*, 3.10 and 2 SNPs were detected in the promoter, exon/intron and 3′UTR regions, respectively (Figure 9B). 

We analyzed the haplotypes of *HvSnRK2.5* in 100 barley core accessions and found that the expression of *HvSnRK2.5* was highly induced in the leaves under drought treatment (Figure 8). Five haplotypes were identified on the basis of the SNPs in the promoter and exon regions (Appendix A). It was found that the barley accessions with *HvSnRK2.5*^Hap3^ kept a higher leaf water content, while the accessions with *HvSnRK2.5*^Hap4^ are quite sensitive to drought stress (Figure 9c). Therefore, it may be suggested that the sequence variation of *SnRKs* affects their expression and response to abiotic stresses.

## 3. Discussion

In this research, 46 *HvSnRK* genes classified into three subfamilies were discovered in the barley genome. The comprehensive studies were performed on the *HvSnRK* family genes, including phylogenetic analysis, gene structures, protein 3D structures, gene distribution on genome, gene duplication and *cis*-elements identification in the promoters. Moreover, utilizing publicly available data, the analysis of *HvSnRKs* expression patterns and SNPs distribution was performed. In our understanding, the current study should be helpful for better understanding the functions of *HvSnRK* genes responding to abiotic stresses.

Previously, 39 *AtSnRK* genes were identified in *Arabidopsis*, including 3 *AtSnRK1s*, 10 *AtSnRK2s* and 26 *AtSnRK3s* [6,32,33,34]; 48 *OsSnRK* genes in *Oryza sativa,* including 4 *OsSnRK1s*, 10 *OsSnRK2s* and 33 *OsSnRK3s* [15]; 34 *EgSnRK* genes in *Eucalyptus grandis,* including 2 *EgSnRK1s*, 8 *EgrSnRK2s* and 24 *EgrSnRK3s* [35]; 44 *BdSnRK* genes in *Brachypodium distachyon*, including 3 *BdSnRK1s*, 10 *BdSnRK2s* and 31 *BdSnRK3s* [36]; and 114 *BnSnRK* genes in *Brassica napus*, including 10 *BnSnRK1s*, 31 *BnSnRK2s* and 73 *BnSnRK3s* [37], respectively. In the barley genome, 46 *HvSnRK* genes classified into 3 subfamilies were discovered, including 4 *HvSnRK1*s, 11 *HvSnRK2*s and 31 *HvSnRK3*s. Furthermore, the phylogenetic tree of the AtSnRK, OsSnRK and HvSnRK proteins showed that the HvSnRKs were closer to the OsSnRKs than AtSnRKs, as expected. 

In *AtSnRK*, *OsSnRK* and *HvSnRK* gene families, various subfamily genes showed the large difference in length and exon-intron structure. In barley, all genes of *HvSnRK1* subfamily contain 11 exons, while 3 of 4 *OsSnRK1* genes have 11exons. All *HvSnRK2* subfamily genes contain eight or nine exons except for *HvSnRK2*.6, which contains three exons due to intron deletion during the evolution process, being consistent with *SnRK2s* in *Arabidopsis*, rice, maize and sorghum [38,39]. The *SnRK3* subfamily can be divided into 2 subgroups, *SnRK3-1*, with more than 10 exons, and *SnRK3-2*, with fewer than 4 exons. The similarity of exon number in the same subfamily may be attributed to their close evolutionary relationship. 

The *cis*-elements in the promoter can regulate gene expression during plant development and the process in response to environmental changes [40,41]. In this study, the promoter sequence analysis showed that various types of *cis*-elements were contained in *HvSnRKS*, such as MBS, ABRE and LTR. The mutations in the MBS element of *ZmSOP2* gene cause the attenuated expression under ABA and dehydration condition in *Zea Mays* [42]. The ABRE element of the *HAHB4* gene was identified to be responsible for ABA, NaCl and drought regulation when the site-directed mutagenesis on the promoter of *HAHB4* was performed on *Arabidopsis* and sunflower [43]. Low temperature response element LTR played a key role in the regulation of *BLT4.9* gene under low temperature stress in barley [44]. We found that most *HvSnRK* genes have at least one of the above *cis*-elements in the promoters. 

In addition, 3D structure modeling was constructed to characterize the potential functions of different HvSnRK subfamilies. A noteworthy characteristic of HvSnRK2 is the well-ordered SnRK2 box, which forms a single α-helix and is packed parallel against the αC-helix in the N-terminal lobe. The SnRK2 box–αC interaction has been reported to be crucial for kinase activity [45]. SnRK3 has a junction region in the C terminal, which packs against αB and αC helices, and the central β-sheet was reported to connect the catalytic domain and the regulatory domain [46]. These modeled 3D structures of HvSnRK proteins lay the foundation for us to understand their biological functions.

Gene expression profiles may provide valuable clues for characterizing gene functions. Here, we analyzed the expression levels of *HvSnRKs* in different barley tissues using published transcriptome data. The findings revealed that the expression patterns of 46 genes could be classified into three groups. (Appendix A). Interestingly, we found that the *HvSnRK*s in group-3 contained fewer *cis*-elements in their promoters than the other two groups. On average, groups-1, -2 and -3 contain 3.94, 4.29 and 2.69 ABRE, respectively. This indicates that *HvSnRK*s activities may be correlated with the disparities of *cis*-elements in the promoter regions. 

In this study, expression patterns of *HvSnRKs* in response to drought, salinity and temperature stresses were also investigated. *HvSnRK3.12* is an orthologous gene of *OsCIPK12* (Figure 5B, Appendix A), which regulates the synthesis of proline and soluble sugar in rice exposed to drought stress [47], indicating the identical functions in response to drought stress. However, *HvSnRK2.5*, which was also significantly induced under drought stress, does not have the orthologous gene in rice. According to the predicted PPI networks (Figure 8), *HvSnRK2.5* may respond to drought stress through regulating ABA signaling. Moreover, we determined the SNPs in each *HvSnRK* gene using the re-sequenced data of a barley core collection. As a result, both potential drought-tolerant and sensitive haplotypes were identified in *HvSnRK2.5*. For example, barley accessions with *HvSnRK2.5*^Hap4^ are tolerant to drought stress, while those accessions with *HvSnRK2.5*^Hap4^ are sensitive (Fig9C). Obviously, these drought-tolerant haplotypes could be used as the elite alleles in drought-tolerant breeding. 

In conclusion, the current study provides a comprehensive understanding of the *SnRK* family genes in barley. Although phylogenetic relation, structure, expression pattern and sequence variation in *HvSnRK* genes were analyzed, further studies are required to reveal the roles of *HvSnRK* family genes in responding to various stresses.

## 4. Materials and Methods

### 4.1. Identification of SnRK Family Proteins in Barley

The amino acid sequences of SnRK proteins from *Arabidopsis thaliana* and rice obtained from the databases TAIR (https://www.arabidopsis.org/ (accessed on 17 August 2020)) and Ensemble-Plants (https://plants.ensembl.org/ (accessed on 17 August 2021)) were used as query probes to search for barley SnRKs using BLASTP against the the IBSC_v2 database from the EnsemblePlants (https://plants.ensembl.org/ (accessed on 17 August 2021)). Meanwhile, using the Hidden Markov Model (HMM) files of SnRKs acquired from the Pfam database (http://pfam.xfam.org/ (accessed on 27 August 2021)), HMMER 3.0 was used to search the SnRKs from the barley protein sequence dataset. All candidate sequences of the SnRKs were confirmed through the NCBI conserved domain database (https://www.ncbi.nlm.nih.gov/cdd (accessed on 27 August 2021)) [48], the SMART database (http://smart.embl-heidelberg.de/ (accessed on 27 August 2021)) [49] and the Pfam database (http://pfam.xfam.org/ (accessed on 27 August 2021)) [50]. Then, the candidate sequences were submitted to the ExPASy (http://www.expasy.ch/tools/pi_tool.html (accessed on 28 August 2021)) to analyze physical and chemical properties and isoelectric points of these proteins. Their subcellular locations were predicted by ProtComp v.9.0 (http://linux1.softberry.com/berry.phtml (accessed on 28 August 2021)).

### 4.2. Analysis of Protein Motifs and Gene Structures 

The amino acid sequences of the SnRKs in barley were uploaded to the MEME online tool (http://meme.sdsc.edu/meme/itro.html (accessed on 29 August 2021)) [51] to analyze the conserved motifs. The settings were selected based on numerous analyses’ experience value: the motif length was set to 6–100 bp, and the maximum number of motifs was set to 10. InterProScan (http://www.ebi.ac.uk/Tools/pfa/iprscan/ (accessed on 29 August 2021)) was used to annotate the motifs. The sequences of barley SnRK genes were extracted from EnsemblePlants database (http://plants.ensembl.org/index.html (accessed on 17 August 2021)), and the gene structure map was created using the Tbtools software [52]. The upstream (−2 kb) sequences of the promoters of *HvSnRK* genes were submitted to the PlantCARE database (http://bioinformatics.psb.ugent.be/webtools/plantcare/html/ (accessed on 29 August 2021)) [53] to analyze the *cis*-acting elements related to abiotic stress such as drought response components MBS (CAACTG) and low temperature response components LTR (CCGAAA).

### 4.3. Phylogenetic Analysis of SnRK Family Proteins in Barley

The full-length SnRK protein sequences of *Arabidopsis*, rice and barley were aligned with Clustal X for multiple sequence alignment (MSA) [54], and 1000 bootstrap tests were performed with MEGA-X software to construct the phylogenetic tree by the Neighbor-Joining (NJ) method [55].

### 4.4. Modeling of 3D Structures of HvSnRKs

The 3D structures of HvSnRK proteins were predicted by homology modeling method. Firstly, the most similar homology of each HvSnRK protein were found in the PDB database (http://www.rcsb.org/ (accessed on 30 August 2021)) by the position-specific iterated BLAST algorithm (PSI-BLAST) [56], then the 3D structures of HvSnRK proteins were predicted by the Swiss-Model interactive tool (https://swissmodel.expasy.org/interactive/ (accessed on 30 August 2021)) [57]. In addition, the 3D structure quality of the HvSnRK proteins was tested by SAVES server (http://nihserver.mbi.ucla.edu/SAVES/ (accessed on 30 August 2021)). Finally, the 3D structures were displayed by the Pymol software.

### 4.5. Chromosomal Location and Gene Duplication of HvSnRKs

The chromosomal location data of *HvSnRKs* was retrieved from the EnsemblePlants, and their distribution was mapped to 7 barley chromosomes using the MapChart v. 2.3 software [58]. *SnRK* duplication patterns, including segmental and tandem duplications, were classified using the Multiple Collinearity Scan toolkit (MCScanX) software [59]. A tandem duplication was defined as a chromosomal area of less than 200 kb containing two or more genes with more than 70% sequence identity between them [60]. Moreover, the synteny relationships of *SnRK* genes between *H. vulgare* and *O. sativa* were performed with python. Non-synonymous (*Ka*) and synonymous (*Ks*) substitutions of each duplicated SnRK gene pair were calculated using the KaKs _Calculator 2.0 [61].

### 4.6. Expression Patterns and Interaction Networks Analysis of HvSnRKs 

The transcriptome data of 16 different tissues of barley cultivar Morex were obtained from the barley transcriptome database (https://apex.ipk-gatersleben.de/apex/f?p=284 (accessed on 3 October 2021)), and the FPKM (Fragments Per Kilobase of exon model per Million mapped fragments) values of *HvSnRKs* were normalized by log2 for further analysis. In addition, barley transcriptome data of drought [28], salt [29], heat [30] and waterlogging [31] treatments were obtained from published references. The heatmaps were performed by the TBtools software. Moreover, the interaction networks of barley SnRK genes were identified based on the database STRING (http://string-db.org/cgi (accessed on 3 October 2021)) [62] and the predicted interaction networks were created with the Cytoscape software [63].

### 4.7. Sequence Variation and Haplotypes of SnRKs in a Barley Core Collection

SNPs in the coding regions of the *HvSnRK* genes were identified in 100 barley accessions from the International Barley Core Selected Collection according to previously re-sequenced data (Appendix A) [64]. For further analysis, SNPs with Minor Allele Frequencies (MAF) more than 5% and a missing rate of less than 20% were employed. Moreover, we performed a soil-culture experiment using the core collections to investigate the relationship between the haplotypes of *SnRK* genes and their drought tolerance. Barley seeds were sowed and grown in the mixture of peat soil and vermiculite with the ratio of 2:1, and watering was stopped for 2 weeks when seedlings were at the 3-leaf stage. The drought tolerance of barley accessions was evaluated based on water content of shoots. Then, we compared shoot water content in each haplotype of the *HvSnRK* genes.

## 5. Conclusions

In this study, 46 *SnRK* family genes were found and classified into 3 subfamilies in barley. Analysis on phylogenetic tree, conserved motifs, gene structure and protein 3D structure of these *SnRK* family genes were performed. The expression profiles showed that the *HvSnRK* genes differed greatly in their expression patterns in 15 barley tissues and in response to various abiotic stresses. The expression of *HvSnRK2.5* and *HvSnRK3.24* were significantly upregulated under drought stress. A protein–protein interaction analysis showed that HvSnRK proteins probably responded to abiotic stresses through ABA signaling. Moreover, the SNPs in each *HvSnRK* gene were detected, and the haplotypes showing drought tolerance and sensitivity were identified in some genes. The current results could be helpful for further exploration of the *HvSnRK* genes associated with abiotic stress tolerance in barley.

## Figures and Tables

**Figure 1 plants-11-00975-f001:**
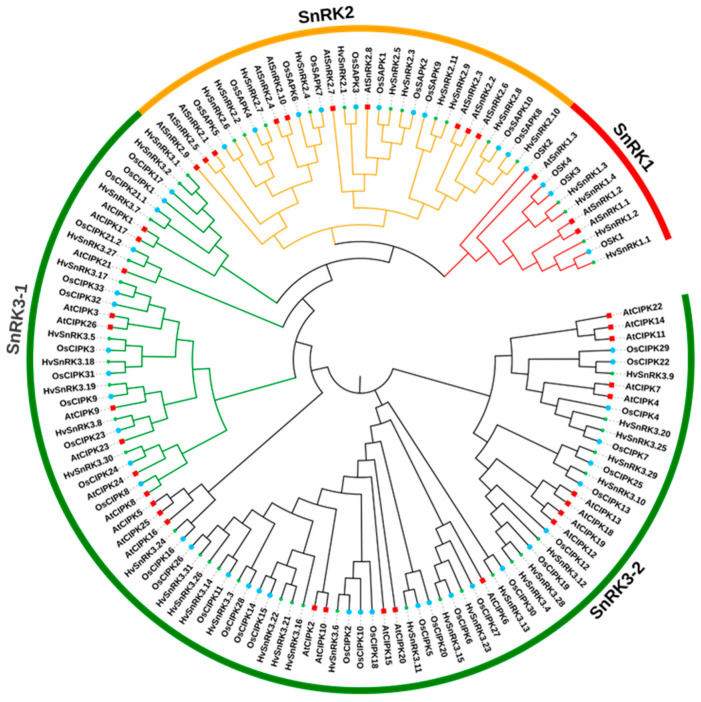
Phylogenetic analysis of SnRK proteins in *O. sativa*, *A. thaliana* and *H. vulgare*. The varied colored arcs represent SnRK protein subfamilies. The 133 SnRK proteins were used to create an unrooted Neighbor-Joining (NJ) phylogenetic tree using MEGA X. The SnRK family proteins from *O. sativa*, *A. thaliana* and *H. vulgare* are represented by the circle, square and star, respectively.

**Figure 2 plants-11-00975-f002:**
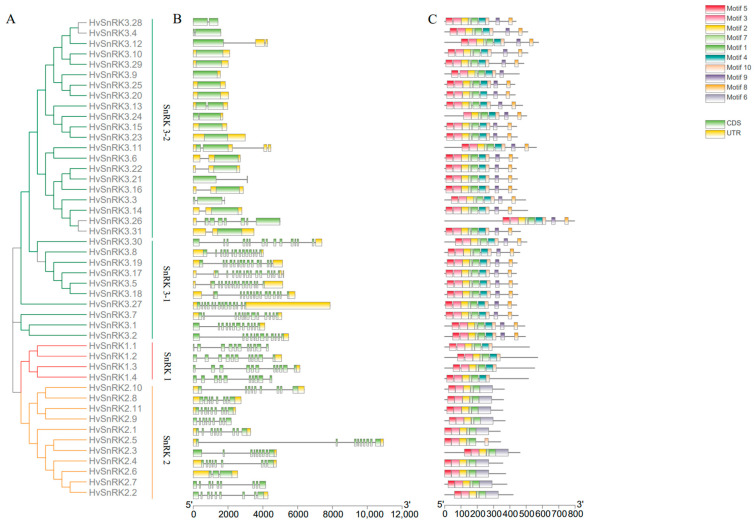
Phylogenetic relationships, gene structures and conserved motifs of the SnRK proteins in *H. vulgare*: (**A**) Phylogenetic tree of 46 HvSnRK proteins; (**B**) Gene structures of *HvSnRK* genes. Green boxes: exons. Black lines: introns. Yellow boxes: UTR areas; (**C**) The motif analysis of HvSnRK proteins. The details of the motifs are provided in Appendix A.

**Figure 3 plants-11-00975-f003:**
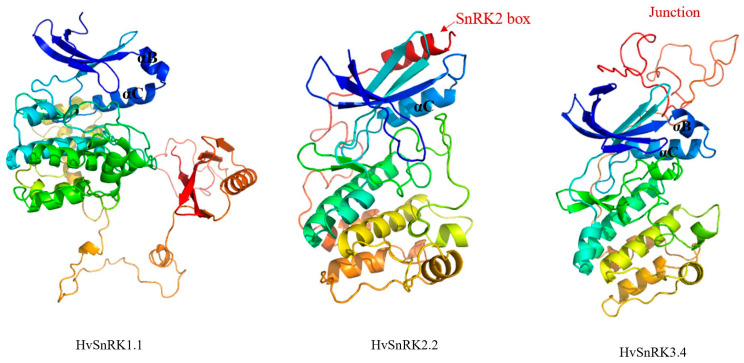
The 3D structure modeling of HvSnRK proteins. The pymol software was used to create the structural image.

**Figure 4 plants-11-00975-f004:**
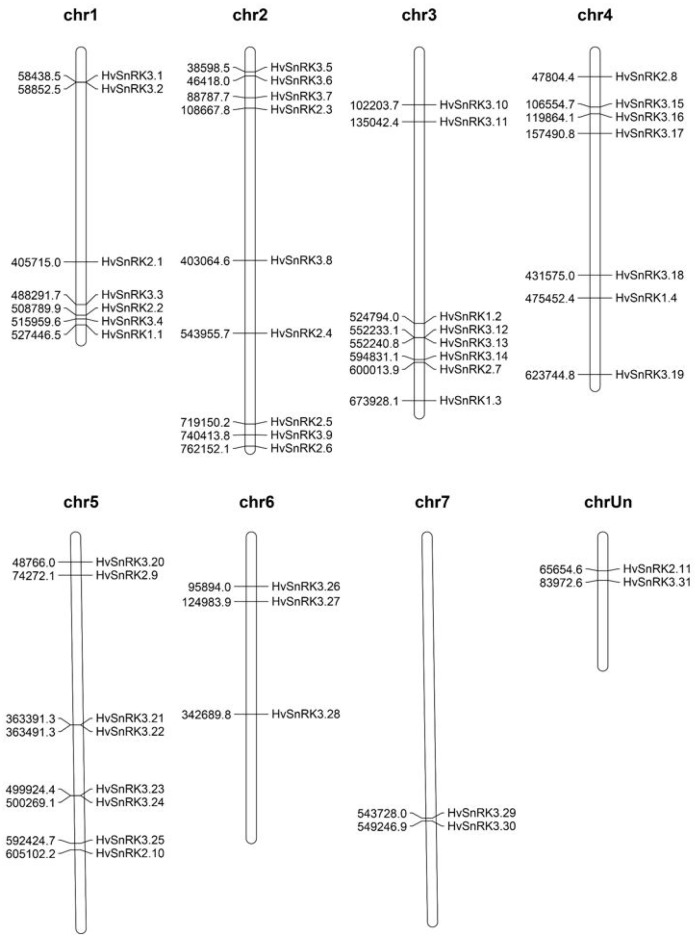
Distribution of *SnRK* genes on the *Hordeum vulgare* genome. The left numbers represent physical location on chromosomes of *HvSnRKs*.

**Figure 5 plants-11-00975-f005:**
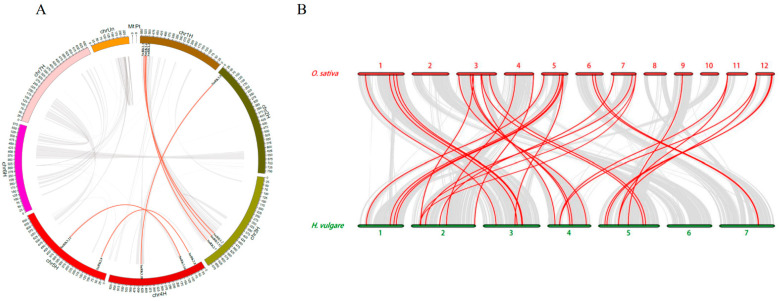
The synteny analysis of *HvSnRK* genes. (**A**) The synteny analysis of *HvSnRK* genes in barley genome. Gray lines: all synteny blocks in the *H.vulgare* genome. Red lines: duplicated *HvSnRK* gene pairs. (**B**) Synteny analysis of *SnRK* genes between the genomes of *O. sativa* and *H. vulgare*. Gray lines: all collinear blocks within *O. sativa* and *H. vulgare*. Red lines: the synteny of *SnRK* genes between *O. sativa* and *H. vulgare*.

**Figure 6 plants-11-00975-f006:**
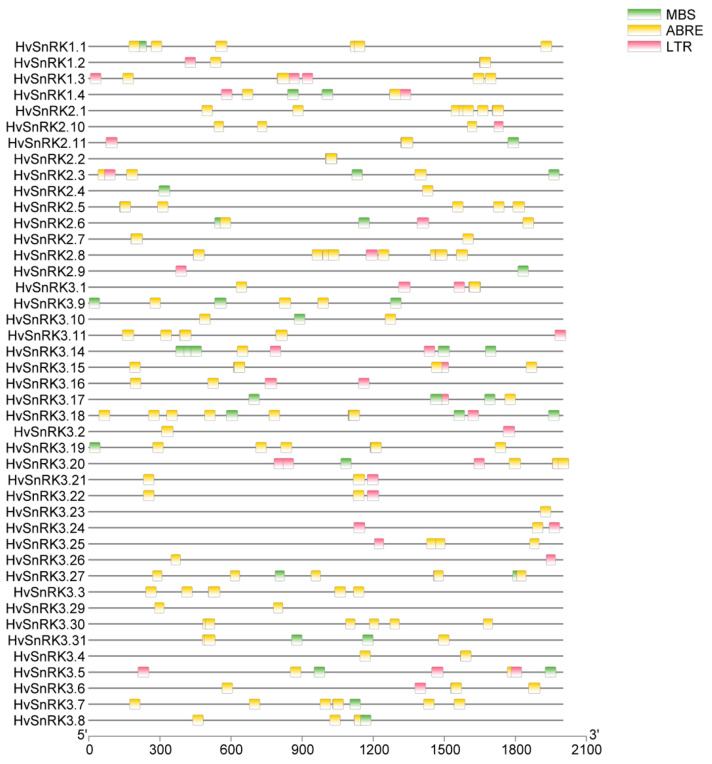
Cis-regulatory elements analysis of *HvSnRK* genes. The green-, yellow- and red-colored boxes indicate MBS, ABRE and LTR *cis*-acting elements, respectively.

**Figure 7 plants-11-00975-f007:**
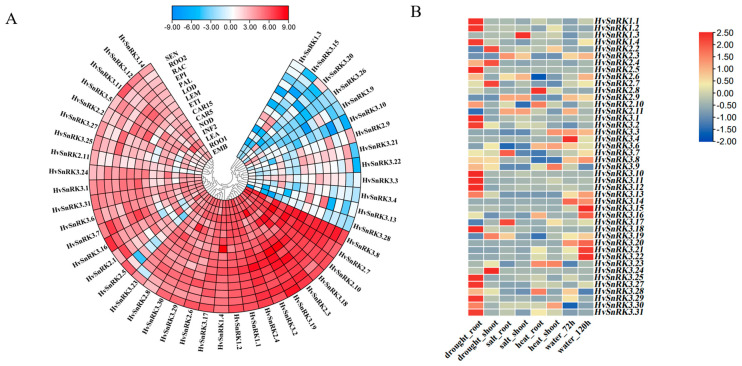
Expression profiles of *HvSnRK* genes. (**A**) Transcriptional profiles of the *HvSnRK* genes in barley different tissues. (**B**) Transcriptional profiles of the *HvSnRK* genes under abiotic stresses. The color scale represents expression data with row scale. Blue: Low expression; Red: High expression.

**Figure 8 plants-11-00975-f008:**
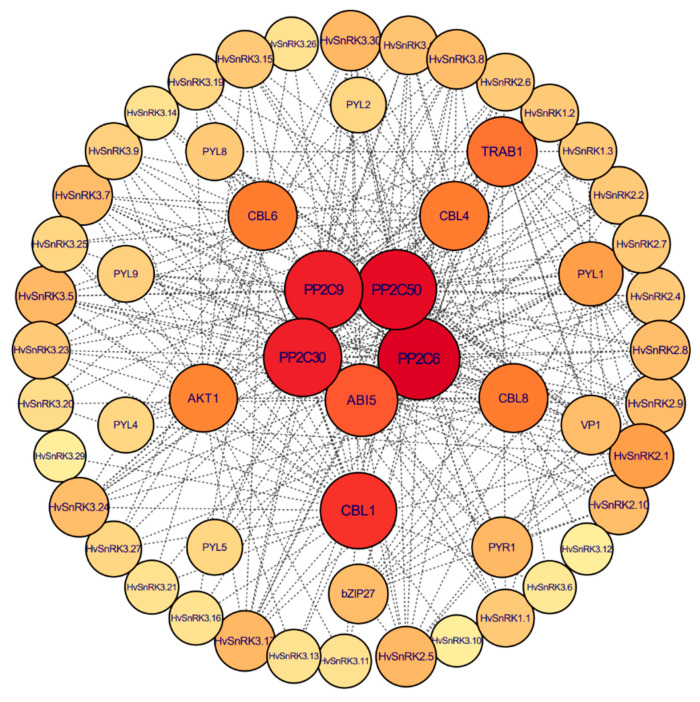
Predicted protein–protein interaction networks of HvSnRK proteins. The external circle represents barley SnRK proteins and the inside circles represent proteins that interact with HvSnRKs. Different layers represent phosphatase 2C family proteins, ion channel protein and PYR/PYL proteins or other transcription factor, respectively. The interaction between these proteins was represented by the gray lines which connected the four circles.

**Figure 9 plants-11-00975-f009:**
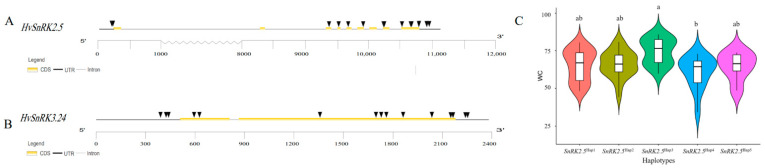
SNPs analysis of *HvSnRK2.5* and *HvSnRK3.24* in 100 accessions of barley core collections. (**A**) Gene structures and SNPs location of *HvSnRK2.5*. (**B**) Gene structures and SNPs location of *HvSnRK 3.24.* Black triangles indicate the SNP distribution. (**C**) Water content of different haplotypes of *HvSnRK2.5* in 100 barley accessions under drought treatment. Capital letters represent significant differences according to Tukey’s honest significant difference (HSD) test (*p* < 0.05).

## Data Availability

Not applicable.

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
