# Peer review of "Genome-Wide Identification, Expression Pattern and Sequence Variation Analysis of SnRK Family Genes in Barley"

_plants, 2022, doi:10.3390/plants11070975_

Round 1

Reviewer 1 Report

Following corrections are required

  1. Lines 159: homeopathic is not the right word to use.
  2. Improve the quality of figure 8, some of the nodes are not clearly visible 

Author Response

Q1. Lines 159: homeopathic is not the right word to use.
A1: Thanks for the suggestion, we have changed "homeopathic elements" to "cis-elements" (L157).

Q2. Improve the quality of figure 8, some of the nodes are not clearly visible
A2: Accordingly, we adjusted the font size of the nods, and improved the quality of figure 8. 

Reviewer 2 Report

Research questions are well defined and within the aims and the scope of the journal. Materials and Methods are mainly properly described and used in a way that is possible to replicate. Otherwise the investigation is mainly performed to good technical standards. It is no ethical problem involved.

Suggestions:

Introduction. Shortly describe the evolutionary significance of duplicated genes.

Line 192. Correct: Functions

Lines 309-3010. Improve the style/grammar.

Material: Define in better way barley material used.

Author Response

Q1. Introduction. Shortly describe the evolutionary significance of duplicated genes. 
A1: Thanks for your valuable comment, and we added some information in the revised MS (L75-79). 

Q2. Line 192. Correct: Functions 
A2: Sorry for our negligence and thanks for the suggestion. We have already made the revision (L189).

Q3. Lines 309-3010. Improve the style/grammar.
A3: We revised the sentence (L305-306) .

Q4. Material: Define in better way barley material used.
A4: Thanks for the suggestion, we have introduced the organization where the barley core accessions are from and listed the detailed information of the barley material in the Table S11. (L369-370)